# Epitaxial Lateral Overgrowth of GaN on a Laser-Patterned Graphene Mask

**DOI:** 10.3390/nano13040784

**Published:** 2023-02-20

**Authors:** Arūnas Kadys, Jūras Mickevičius, Kazimieras Badokas, Simonas Strumskis, Egidijus Vanagas, Žydrūnas Podlipskas, Ilja Ignatjev, Tadas Malinauskas

**Affiliations:** 1Institute of Photonics and Nanotechnology, Vilnius University, Sauletekio Ave. 3, LT-10257 Vilnius, Lithuania; 2Evana Technologies, Ltd., Mokslininku St. 2A-120, LT-08412 Vilnius, Lithuania; 3Center for Physical Sciences and Technology, Sauletekio Ave. 3, LT-10257 Vilnius, Lithuania

**Keywords:** GaN, MOCVD, ELO, graphene, laser ablation, cathodoluminescence

## Abstract

Epitaxial lateral overgrowth (ELO) of GaN epilayers on a sapphire substrate was studied by using a laser-patterned graphene interlayer. Monolayer graphene was transferred onto the sapphire substrate using a wet transfer technique, and its quality was confirmed by Raman spectroscopy. The graphene layer was ablated using a femtosecond laser, which produced well-defined patterns without damaging the underlying sapphire substrate. Different types of patterns were produced for ELO of GaN epilayers: stripe patterns were ablated along the ${\left[\bar{1}100\right]}_{\mathrm{sapphire}}$ and ${\left[11\bar{2}0\right]}_{\mathrm{sapphire}}\;$ directions, a square island pattern was ablated additionally. The impact of the graphene pattern on GaN nucleation was analyzed by scanning electron microscopy. The structural quality of GaN epilayers was studied by cathodoluminescence. The investigation shows that the laser-ablated graphene can be integrated into the III-nitride growth process to improve crystal quality.

## 1. Introduction

Lack of native substrate has always plagued the development of III-nitride materials and devices. GaN-based structures are usually deposited on various foreign substrates, such as sapphire or SiC; however, the large lattice and thermal mismatch causes high dislocation densities [1,2]. To improve the material quality, various technological approaches have been utilized, such as a multi-step growth procedure [2,3], the introduction of various interlayers to the structure [4,5], and the pulsed growth [6,7]. Recently, another promising approach has emerged, based on graphene as an interlayer between the substrate and the epitaxial layer [8,9]. The weak van der Waals bond at the epilayer/graphene interface relaxes the lattice and thermal mismatch [10], while keeping the epilayer aligned to the substrate below the graphene interlayer [11]. On the other hand, the nucleation on the pristine graphene is very difficult due to the absence of dangling bonds on its surface [12], and GaN island growth initiates at the graphene defect sites [13]. Then again, if graphene possesses defects, such as holes, it can act as a mask rather than a proper interlayer, and the epitaxial lateral overgrowth (ELO) of GaN utilizing self-organized graphene as a mask has attracted attention [14,15]. Achieving accurate control and high reproducibility of graphene masks requires reliable patterning techniques, such as lithography. An attractive possibility to pattern graphene lies in using the ultrafast lasers [16,17]. Being contactless, this method reduces undesired modification of graphene, while due to its limited thermal influence, it can be applied for patterning graphene on sensitive substrates. Graphene can be used as transparent contact in the GaN technology [18,19]; thus, the integration of laser-ablated graphene into the GaN growth process is in high demand. 

In this work, we present graphene ablation on sapphire substrates with a femtosecond laser and evaluate if the resulting patterns are suitable for ELO of GaN epilayers. Monolayer graphene is transferred onto the sapphire substrate using a wet method; its quality is assessed by Raman spectroscopy. The laser-ablated patterned structures are analyzed by scanning electron microscopy (SEM). Afterward, the patterned structures are used for the growth of GaN layers, and the structural properties of the epilayers are studied.

## 2. Materials and Methods

Epi-ready 2-inch monocrystal (0001) orientation sapphire substrates were used for graphene layer transfer and further deposition of GaN epilayers. Commercially available transfer-ready poly(methyl methacrylate)-coated (PMMA-coated) graphene monolayers from Graphenea Inc. (San Sebastian, Spain) were used to cover the sapphire substrates. Rapid thermal annealing (RTA) was used for the thermal treatment of the samples. The GaN epilayers were deposited by low-pressure metalorganic vapor phase epitaxy (MOVPE) using a flip-top close-coupled showerhead 3×2” reactor (AIXTRON, Herzogenrath, Germany). Trimethylgallium (TMGa) and ammonia (NH_3_) were used as Ga and N precursors, respectively. The epitaxy process was monitored by an in situ laser reflectometry system operating at 650 nm. 

Raman measurements for graphene characterization were performed using a confocal Raman microscope (Renishaw, Wotton-under Edge, UK). The 532 nm laser excitation source with a power of 1.8 mW was focused on a 0.9 µm diameter spot on the sample surface, and 1800 lines/mm grating was used to record the Raman spectra. The wavenumber axis was calibrated using a polystyrene standard. Femtosecond laser microfabrication setup (Evana Technologies, Vilnius, Lithuania) based on Carbide laser (Light Conversion, Vilnius, Lithuania) and Aerotech xy stages were used to perform laser ablation. The surface of the samples after ablation and epitaxy experiments was studied by SEM (CamScan Apollo 300, Cambridge, United Kingdom, now successor Applied Beams, LLC, Beaverton, OR, USA). Spatially-resolved cathodoluminescence (CL) measurements were performed using a hybrid CL-SEM system (Attolight Chronos, Lausanne, Switzerland). The electron beam was accelerated to 4 keV; the collected light was dispersed using a spectrometer (Horiba iHR, Kyoto, Japan) and recorded using a CCD camera (Andor Newton, Belfast, United Kingdom). All the measurements were performed at room temperature.

## 3. Results and Discussion

The key processes involved in the sample preparation are schematically illustrated in Figure 1. Four main steps can be identified: (a) transfer of monolayer graphene onto the sapphire substrate by the wet transfer; (b) laser ablation of graphene; (c) nucleation of GaN seeds; and (d) epitaxial lateral overgrowth of GaN film.

### 3.1. Graphene Layer Transfer

Monolayer graphene pieces of size 1.3 × 1.3 cm^2^ were transferred onto the sapphire substrates using the wet transfer procedure. First, the PMMA/graphene cuts were dipped into deionized water and left floating freely on the water surface for 5 min. Afterward, the sapphire substrate was put beneath the floating polymer/graphene piece, which was attached as close as possible to the center of the substrate. The sample was then dried in the air until there were no water droplets visible. Next, the sample was heated inside the RTA oven for 20 min at a temperature of 100 °C under an N_2_ atmosphere. The PMMA was dissolved by dipping the sample into acetone for 45 min at 40 °C and rinsing in isopropyl alcohol for another 45 min at 40 °C. Finally, the samples were annealed for 8 h using RTA at 300 °C in a vacuum to remove any organic surface residues left [20].

The quality of transferred graphene on the sapphire substrate was verified by Raman spectroscopy. The obtained Raman spectrum, shown in Figure 2, exhibits the graphene fingerprint modes G and 2D. Defect-related D mode also manifested itself at around 1355 cm^−1^. However, the intensity of D mode was low compared to G (I_G_/I_D_ approx. 12), indicating the high quality of the graphene [21,22]. The ratio of 2D and G peak intensities (I_2D_/I_G_) was 3.8, while the position and full width at half maximum (FWHM) of the 2D peak were 2683 cm^−1^ and 44 cm^−1^, respectively, indicating monolayer graphene [23].

### 3.2. Graphene Lithography

Laser processing of the graphene layer was performed using a 1030 nm laser line with a repetition rate of 60 kHz and 350 fs pulse duration; the pulse energy was set to 60 nJ. The laser beam was focused onto the sample by means of an aspherical objective lens with a focal length of 4 mm; the ablated areas were scribed with a scanning speed of *v* = 6 mm/s. To check the sensitivity of the experimental setup to surface roughness, the initial ablation experiments were performed without using the surface tracking system. Figure 3 presents the optical image of the graphene layer ablated by varying the distance between the focusing lens and the sample surface. Two types of surface modification can be observed: the wide gray stripes of ~4 µm width indicate the removal of graphene, and the narrow dark lines with the width below 1 µm are due to the modified surface of the sapphire substrate. The laser fluence at the center of the Gaussian beam at the focal plane was ~9 J/cm^2^, which was significantly above the ablation threshold of both graphene (~70 mJ/cm^2^ [16]) and sapphire (~2 J/cm^2^ [24]). Changing the distance between the focusing lens and the sample surface led to laser beam defocusing and a larger spot diameter, which, in turn, produced lower laser fluence. The shift of the focal plane by a couple of microns already resulted in laser fluence below the sapphire ablation threshold when only the graphene layer was affected (see Figure 3). The strong dependence of the damage profile on the distance between the focusing lens and the sample surface occurred due to the distance variation of 1 µm being similar to the depth of focus of our system (~2.5 µm). Therefore, a tracking system autocorrecting for all the surface irregularities was necessary for the reproducible ablation process.

The subsequent ablation experiments were performed employing the surface tracking system. To remove 2-µm-wide stripes of graphene without damaging the sapphire substrate, the focal plane was set at the 5 µm distance from the sample surface, and the pulse energy was reduced. The 2 µm width of the graphene window was selected to ensure fast island coalescence during the ELO process: the diameter of the GaN islands reaches a couple of microns before fully merging into a 2D layer [25]. 

Several patterns with different distances between graphene windows were fabricated, as shown in SEM images in Figure 4. The pattern orientation was along the ${\left[\bar{1}100\right]}_{\mathrm{sapphire}}$ direction of the sapphire substrate. The areas without graphene revealed undamaged sapphire, even though some ablation debris could be spotted. The thinnest obtained graphene stripes were 200 nm wide; however, the stripe width was highly uneven [see Figure 4a]. The evenness slightly improved for thicker stripes with the standard deviation of stripe width of 50 nm. Nevertheless, the overall pattern showed a nice periodic structure with repeating stripes.

For the following ELO of GaN epilayers, the graphene layer was divided into four quadrants (zones A-D, as illustrated in Figure 5), and periodic patterns were fabricated in three of them. The periodic structure consisting of alternating graphene stripes and windows, both having the width of 2 µm and running along the ${\left[\bar{1}100\right]}_{\mathrm{sapphire}}\;$ direction was fabricated in zones A and B. Analogous periodic structure but running along the ${\left[11\bar{2}0\right]}_{\mathrm{sapphire}}\;$ direction was fabricated in zones B and C. Since both structures intersected in zone B, it resulted in square graphene islands with a side length of 2 µm. A single quadrant (zone D) was kept with unmodified graphene as a reference. The obtained graphene pattern is schematically illustrated in Figure 5.

### 3.3. ELO of GaN Epilayers

The GaN epilayers were deposited on top of the laser-patterned graphene by MOVPE. First, we evaluated the suitability of the sapphire substrate covered by different laser-ablated graphene patterns for the formation of GaN seeds. The epitaxial process in the reactor started from the in situ surface cleaning procedure in the H_2_ atmosphere and reactor pressure of 150 mBar. To preserve the graphene layer, the cleaning temperature was reduced from the standard 1100 °C down to 800 °C: it is still high enough to remove the possible organic and water contamination, although it is too low to modify the sapphire surface by removing some oxygen [26]. The surface was nitridated at 575 °C and 600 mBar for 140 s. Following the nitridation, a low-temperature (LT) GaN layer was deposited for 250 s at 575 °C using flow rates of TMGa and NH_3_ at 0.05 µmol/min and 80 µmol/min, respectively. The thickness of the LT-GaN layer was ~40 nm. The recrystallization and seed formation was performed at 1100 °C and 400 mBar. To increase the seed size, extra growth was performed for 180 s using flow rates of TMGa and NH_3_ at 0.16 µmol/min and 130 µmol/min, respectively. All steps were carried out in the H_2_ atmosphere.

The formation of GaN seeds on different surfaces is shown in the SEM images in Figure 5. It is evident that GaN seeds are formed only on sapphire (either bare or in the graphene windows), with practically no seeds formed on graphene. Such distribution is related to the different GaN nucleation processes on sapphire and graphene. GaN nucleation on sapphire is mainly due to the chemical adsorption process, while physical adsorption dominates in nucleation on graphene due to its low surface energy [12]. Nucleation on graphene is enhanced in the remote epitaxy, which occurs due to the interaction between the sapphire substrate and GaN through a monolayer graphene [11,27]; however, it is still significantly less effective compared to chemical adsorption. 

Meanwhile, the concentration of the dominant seeds with diameter larger than 0.5 µm is very similar both on bare sapphire and on the surface with laser-ablated graphene pattern, regardless of the pattern details. This points to a clean laser ablation process with no damage to the sapphire epi-ready surface or possible contamination by carbon.

Following the seed formation, the growth continued until GaN islands coalesced into a complete 2D layer on the surface of a bare sapphire. The island coalescence was performed at 1080 °C and 150 mBar using flow rates of TMGa and NH3 at 0.16 µmol/min and 112 µmol/min, respectively.

Figure 6 presents the SEM images of the GaN layers grown on different graphene patterns. A nearly continuous 2D layer was formed on the surface with a square graphene island pattern (zone B). The striped graphene pattern resulted in an incomplete GaN layer (zones A and C). Furthermore, a certain difference could be observed for different graphene stripe orientations. Considering that hexagonal GaN lattice has a 30° rotation from sapphire, the relative growth rate of GaN ${\left(11\bar{2}0\right)}_{\mathrm{GaN}}$ plane is higher than that of the ${\left(1\bar{1}00\right)}_{\mathrm{GaN}}$ plane. Similar trends of growth anisotropy have been reported for GaN growth using SiO_2_ or SiN masks both on sapphire and SiC substrates [28,29]. Meanwhile, many small, closely packed islands composed the GaN layer deposited on the unmodified graphene (zone D). The detailed analysis of the surface morphology revealed a flat surface with well-ordered monolayer steps for GaN grown on patterned graphene (zones A, B, and C), as expected for growth on a sapphire substrate with a vicinal angle toward the *m*-axis of 0.2 ÷ 0.4° [30], although some step bunching is visible. On the other hand, no step-flow growth mode could be observed on the surface of GaN islands grown on unmodified graphene (zone D). The surface with macrosteps and islands indicates insufficient large holes in the graphene layer for GaN seed formation. As discussed above, this is due to weak nucleation on graphene.

To compare the structural quality of GaN layers grown on different graphene patterns, dislocation densities (DD) were evaluated in each zone using CL. The typical images of the spatial distribution of integrated CL intensity are shown in Figure 7. Dislocations manifest in the images by spots of low CL intensity (dark spots), with the density of dark spots corresponding to dislocation density. The dark spot capture in CL images was performed using Laplacian of Gaussian blob detection algorithm. 

The highest average DD value of 1.1 × 10^9^ cm^−2^ was obtained for GaN grown on square graphene islands (zone B). Somewhat lower values of 8.1 × 10^8^ and 6.6 × 10^8^ cm^−2^ were estimated for GaN grown on striped graphene pattern (zones C and A, respectively). Meanwhile, the lowest DD values of 5 × 10^8^ cm^−2^ were measured both in GaN deposited on unmodified graphene (zone D) and in reference GaN grown on bare sapphire. The link between graphene pattern and dislocation density can be interpreted by considering the organization of dislocations along the lines parallel to the direction of stripes [29], as indicated by the white dashed lines in Figure 7. Since highly defective lines correspond to the coalescence boundaries, the nearly fully coalesced layer in zone B has the highest DD. On the other hand, the slow lateral growth along the ${\left[1\bar{1}00\right]}_{\mathrm{GaN}}$ direction in zone A causes incomplete island coalescence and results in the lowest DD. The results suggest that the graphene stripe width should be larger for more efficient ELO. Meanwhile, growth on graphene without any patterns could be used for dislocation reduction.

## 4. Conclusions

To summarize: femtosecond laser patterning of a graphene monolayer on a sapphire substrate is developed to fabricate a mask for ELO of GaN epilayers. Applicability of patterned graphene for lateral overgrowth is demonstrated: GaN seeds are formed mostly in the graphene windows during the standard MOVPE growth procedure. Growth utilizing masks comprised of graphene stripes confirmed higher GaN growth rate along the ${\left[11\bar{2}0\right]}_{\mathrm{GaN}}$ direction compared to the ${\left[1\bar{1}00\right]}_{\mathrm{GaN}}$ direction. Cathodoluminescence analysis of dislocations revealed the arrangement of dislocations at the boundaries between coalescing GaN islands formed in different mask windows. The graphene-ELO technology shows the promising application of 2D materials as masks both for improving the crystal quality and for integration into the GaN-based structures.

## Figures and Tables

**Figure 1 nanomaterials-13-00784-f001:**
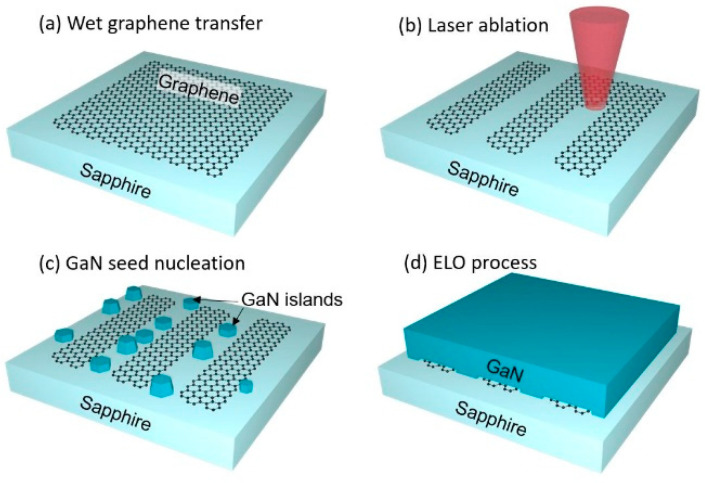
Schematic diagram of the key processes involved in the growth of GaN: (**a**) wet transfer of monolayer graphene; (**b**) laser ablation of graphene; (**c**) nucleation of GaN seeds; and (**d**) ELO of GaN film. The objects are not to scale.

**Figure 2 nanomaterials-13-00784-f002:**
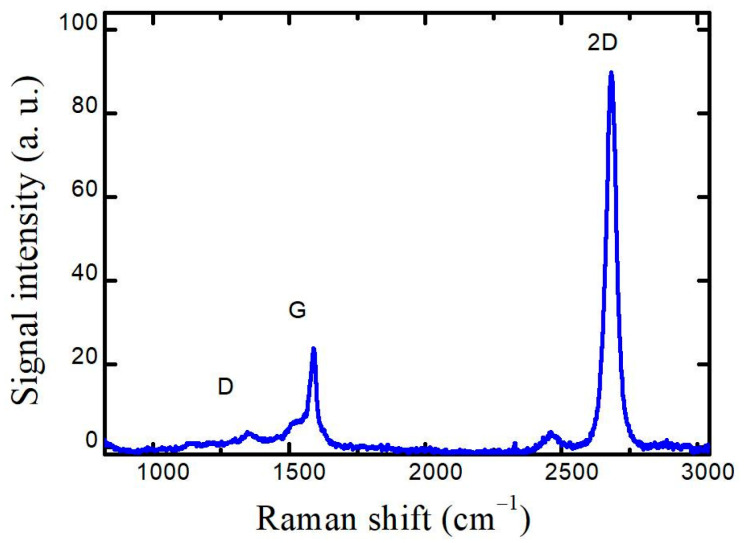
Raman spectrum measured after the transfer of graphene prior to laser patterning. Characteristic graphene modes are indicated.

**Figure 3 nanomaterials-13-00784-f003:**
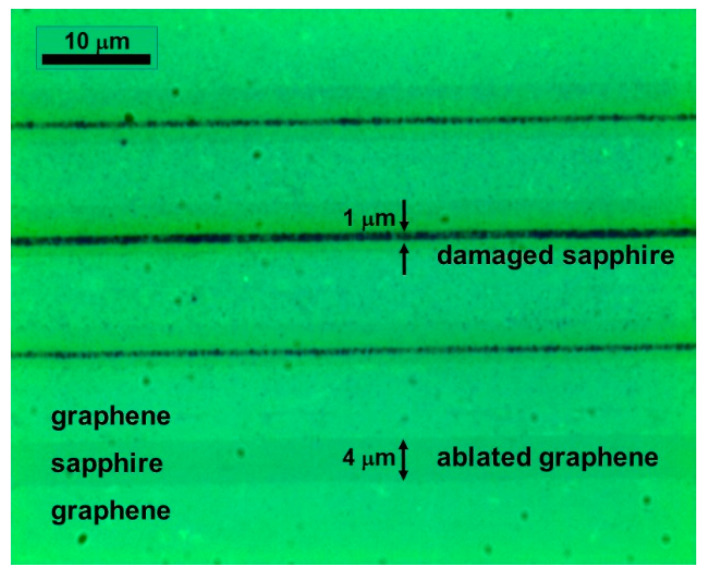
Optical image of laser-ablated graphene, demonstrating the influence of the distance between the focusing lens and the surface of the sample. Distance is varied by the step size of 1 µm.

**Figure 4 nanomaterials-13-00784-f004:**
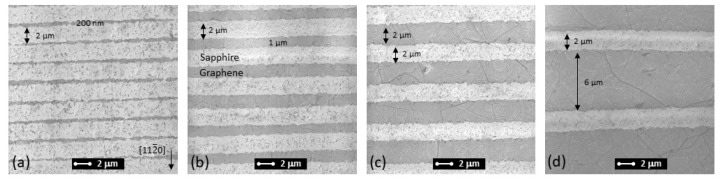
SEM images of laser-ablated graphene patterns on sapphire substrate. The graphene window width is 2 µm, the distance between windows is equal to 200 nm (**a**), 1 µm (**b**), 2 µm (**c**), and 6 µm (**d**). An arrow in (**a**) denotes ${\left[11\bar{2}0\right]}_{\mathrm{sapphire}}$ direction of an underlying sapphire substrate, and is the same for all images.

**Figure 5 nanomaterials-13-00784-f005:**
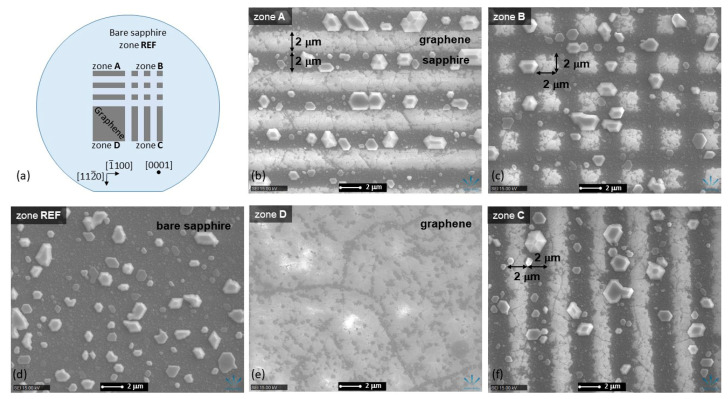
Schematic diagram of graphene pattern used for ELO of GaN epilayers (**a**). SEM images of GaN seeds on each graphene pattern type (**b**,**c**,**f**); unmodified graphene (**e**); and bare sapphire surface (**d**). The directions of sapphire crystal lattice are indicated.

**Figure 6 nanomaterials-13-00784-f006:**
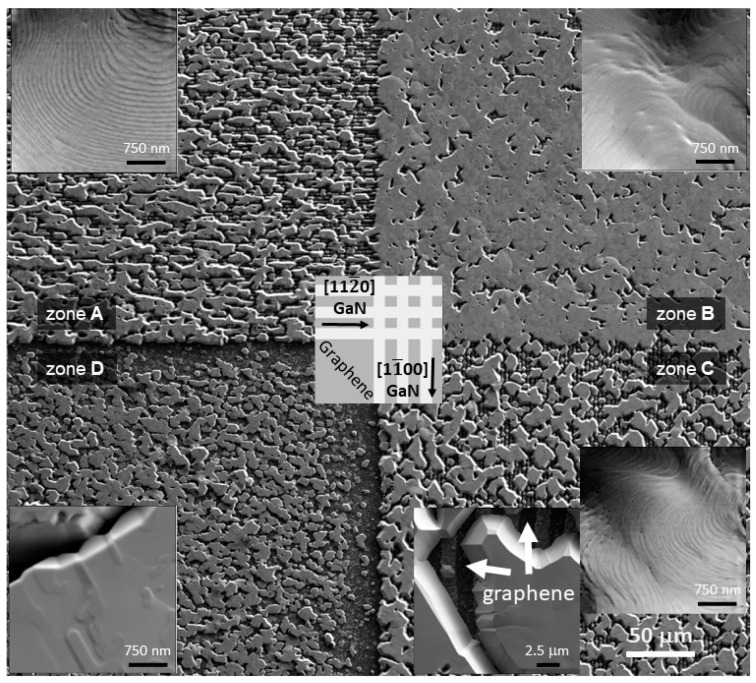
SEM images of GaN layers grown using ELO on different graphene patterns (indicated in the center of the figure, the directions represent GaN crystal lattice).

**Figure 7 nanomaterials-13-00784-f007:**
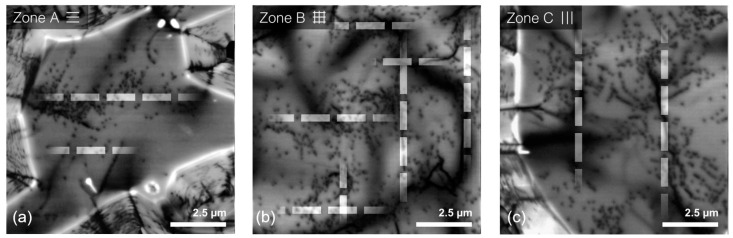
CL intensity images of GaN layers grown using ELO on different graphene patterns (indicated) (**a**–**c**). White dashed lines illustrate the correlation between graphene pattern and agglomerations of dislocations.

## Data Availability

The data that support the findings of this study are available from the corresponding author upon reasonable request.

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
