# Peer review of "Epitaxial Lateral Overgrowth of GaN on a Laser-Patterned Graphene Mask"

_nanomaterials, 2023, doi:10.3390/nano13040784_

Round 1

Reviewer 1 Report

Referee report

The article “Epitaxial lateral overgrowth of GaN on a laser-patterned graphene mask” is devoted to studying of GaN growth on patterned graphene layers. It is quite actual and interesting topic, The authors proposed an original approach based on the ablation of graphene under the action of ultrashort laser pulses. The article contains new experimental data and will be interesting for researchers and technologists. The article can be published after minor revision.

Comment:

1) The work is good in technological terms, but still I would like the authors to at least make assumptions about the physical mechanism of selective growth of GaN. Why does gallium nitride preferentially grow on graphene layers? It is known that sapphire substrates are also used for the growth of gallium nitride. Could it be a matter of modifying the sapphire surface with a laser beam? Maybe there is a partial ablation of sapphire? Then it is very important to know exactly the laser fluence and that will be my next comment. (By the way, graphene is not a polar material, so the problem of twinning during growth can arise. For example, when growing on silicon, the silicon surface is nitridizided in order to avoid the problem of twinning. It would be nice to also raise this issue in the discussion).

2) It is known that lasers have a Gaussian power density distribution depending on the distance to the beam center. In this regard, the determination of laser fluence is a very non-trivial task, see for example the article “Single-shot selective femtosecond and picosecond infrared laser crystallization of an amorphous Ge/Si multilayer stack” Optics and Laser Technology, v.161, 109161 (2023) DOI: https://doi.org/10.1016/j.optlastec.2023.109161and references in it. Did the authors mean the laser fluence at the center of the beam? This part needs to be described in more detail.

3) The authors present the Raman spectrum of graphene, but discuss it very sparsely. It would be possible to analyze the ratio of the G and D peaks, and also try to deconvolute the spectrum into peaks and find the peaks corresponding to defects (D’) and inclusions of amorphous carbon, as, for example, in the works of Nebogatikova - N.A. Nebogatikova, et. al. Nanostructuring few-layer graphene films with swift heavy ions for electronic application: tuning of electronic and transport properties. Nanoscale, v.10, issue 30, pp. 14499-14509, 2018 DOI: 10.1039/c8nr03062f and “Visualization of Swift Ion Tracks in Suspended Local Diamondized Few-Layer Graphene” Materials, 16, 1391 (2023). DOI: https://doi.org/10.3390/ma16041391.

Accept with minor revision.

Author Response

We would like to thank the reviewers for their thoughtful comments and efforts towards improving our manuscript

Comment 1.1: The work is good in technological terms, but still I would like the authors to at least make assumptions about the physical mechanism of selective growth of GaN. Why does gallium nitride preferentially grow on graphene layers? It is known that sapphire substrates are also used for the growth of gallium nitride. Could it be a matter of modifying the sapphire surface with a laser beam? Maybe there is a partial ablation of sapphire? Then it is very important to know exactly the laser fluence and that will be my next comment. (By the way, graphene is not a polar material, so the problem of twinning during growth can arise. For example, when growing on silicon, the silicon surface is nitridizided in order to avoid the problem of twinning. It would be nice to also raise this issue in the discussion).

Answer: We would like to thank the reviewers for their thoughtful comments and efforts towards improving our manuscript. We would like to stress that we have shown that graphene acts as a mask in ELO process, i.e. graphene suppresses the growth of GaN. The formation rate of GaN seeds on graphene is much lower compared to that on sapphire as can be clearly seen in Zone A, B and C in Figure 4. So GaN preferentially does not grow on graphene, rather on sapphire. The graphene has no dangling bonds and thus very low surface energy as was pointed out in a manuscript, thus nucleation of GaN on graphene is less probable. However as was recently shown graphene interlayer does not completely screen the potential field of substrate thus can be used for remote epitaxy (or epitaxy through graphene).

Comment 1.2. It is known that lasers have a Gaussian power density distribution depending on the distance to the beam center. In this regard, the determination of laser fluence is a very non-trivial task, see for example the article “Single-shot selective femtosecond and picosecond infrared laser crystallization of an amorphous Ge/Si multilayer stack” Optics and Laser Technology, v.161, 109161 (2023) DOI: https://doi.org/10.1016/j.optlastec.2023.109161and references in it. Did the authors mean the laser fluence at the center of the beam? This part needs to be described in more detail.

Answer 1.2 Yes, we indeed agree that accurate determination of laser fluence when laser is sharply focused is demanding task. Our number is estimates of laser fluence at the center of Gausian beam, obtained by well-established formula of , where is w is radius at  level. Radius w was calculated by measuring diameter of the beam before focusing. So, we have expanded the description in the manuscript in section 3.2. We did not speculate about exact fluence of the beam after we defocused it.

The laser fluence at the center of Gaussian beam at the focal plane was ~9 J/cm2, which was significantly above the ablation threshold of both graphene (~70 mJ/cm2 [16]) and sapphire (~2 J/cm2 [20]).

Comment 1.3. The authors present the Raman spectrum of graphene, but discuss it very sparsely. It would be possible to analyze the ratio of the G and D peaks, and also try to deconvolute the spectrum into peaks and find the peaks corresponding to defects (D’) and inclusions of amorphous carbon, as, for example, in the works of Nebogatikova - N.A. Nebogatikova, et. al. Nanostructuring few-layer graphene films with swift heavy ions for electronic application: tuning of electronic and transport properties. Nanoscale, v.10, issue 30, pp. 14499-14509, 2018 DOI: 10.1039/c8nr03062f and “Visualization of Swift Ion Tracks in Suspended Local Diamondized Few-Layer Graphene” Materials, 16, 1391 (2023). DOI: https://doi.org/10.3390/ma16041391.

Answer: 1.3. We have expanded our analysis including information about D peak. The low intensity of D peak strengthens our point that we have good quality single layer graphene. We have included this sentence and citations in the manuscript.

“Defect-related D mode also manifested itself at around 1355 cm-1. However, the intensity of D mode was low compared to G (IG/ID approx. 12) indicating high quality of graphene [https://doi.org/10.1038/nnano.2013.46, https://doi.org/10.3390/ma16041391]”.

Reviewer 2 Report

The authors studied epitaxial lateral overgrowth (ELO) of GaN epilayers on sapphire substrate by using laser-patterned graphene interlayer  This work is of value for  improving the crystal quality and for integration into GaN-based structures. The manuscript should be published in Nanomaterials after authors amend some minor items listed below.

1. The significance of the research is not pointed out in abstract, and the highlight of the research is not highlighted. The authors should state the improvement of GaN-based structures performance by this study in abstract.

2. In the introduction, the authors did not quote and sort out enough of the relevant references.

3. The authors need to draw a diagram to illustrate the whole fabrication process of GaN-based structures. From graphene layer transfer to ELO.

4. The authors need to enlarge the scale bar text in SEM images of Figure 4.

Author Response

We would like to thank the reviewers for their thoughtful comments and efforts towards improving our manuscript

Comment 2.1. The significance of the research is not pointed out in abstract, and the highlight of the research is not highlighted. The authors should state the improvement of GaN-based structures performance by this study in abstract.

Answer: Thank you for your comment. We have added a sentence into abstract to highlight the significance of the manuscript.

The investigation shows that the laser ablated graphene can be integrated in III-nitride growth process to improve crystal quality

Comment 2.2. In the introduction, the authors did not quote and sort out enough of the relevant references.

Answer: We added another point why integration of graphene with GaN is important.

Graphene can be used transparent contact in GaN technology [18,19] thus integration of laser ablated graphene into GaN growth process is in high demand.

Comment 2.3. The authors need to draw a diagram to illustrate the whole fabrication process of GaN-based structures. From graphene layer transfer to ELO.

Answer: We have included the Figure into the manuscript illustrating the schematic diagram of the key processes of this work.

Comment 2.4. The authors need to enlarge the scale bar text in SEM images of Figure 4.

Answer. We have enlarged text in the Figure 4.

Reviewer 3 Report

In this work, epitaxial lateral overgrowth of GaN epilayers on sapphire substrates was studied by using laser-patterned graphene interlayer. However, there are still some problems in this paper that need to be revised as follows.

1 The scale is missing in Figure 3 and should be added.

2 The paper repeatedly emphasizes that the sapphire substrate was not damaged in the graphene removal experiments, but direct evidence for this is lacking

3 In Figure 4, the authors show the state of GaN seed formation on the surface of bare sapphire. However, the GaN film formation state on the bare sapphire surface is not shown in the follow-up. This part should be added as a way to determine the effect of the presence or absence of graphene on GaN film formation

4 In Figure 3, the authors show that graphene can be prepared to 200 nm, 1 μm, 2 μm, and 6 μm wide, respectively. The graphene material with a width of 2 μm was used for the subsequent growth experiments. When the dislocation density is calculated, it seems to show that the larger the graphene area is, the smaller the dislocation density is. Therefore, what was the basis for the authors' choice of graphene of 2 μm width?

Author Response

We would like to thank the reviewers for their thoughtful comments and efforts towards improving our manuscript

Comment 3.1 The scale is missing in Figure 3 and should be added.

Answer. We have added the scale bar to the Figure 4.

Comment 3.2 The paper repeatedly emphasizes that the sapphire substrate was not damaged in the graphene removal experiments, but direct evidence for this is lacking

Answer. The direct proof of damaged sapphire can be seen in optical image of sample in Figure 3. We do not see any damage to the sapphire surface when laser fluence was decreased below sapphire ablation threshold. Moreover, epitaxy is very sensitive to surface morphology. Experiments show that GaN seed formation is not altered, which provides indirect proof that epi-ready sapphire surface is not modified by laser.

Comment 3.3 In Figure 4, the authors show the state of GaN seed formation on the surface of bare sapphire. However, the GaN film formation state on the bare sapphire surface is not shown in the follow-up. This part should be added as a way to determine the effect of the presence or absence of graphene on GaN film formation.

Answer. Below we provide SEM image of GaN grown on bare sapphire. It is a standard MOVPE growth proccess of GaN. Our focus is GaN growth over graphene by ELO process, thus we did not include that figure in the manuscript.

Comment 3.4 In Figure 3, the authors show that graphene can be prepared to 200 nm, 1 μm, 2 μm, and 6 μm wide, respectively. The graphene material with a width of 2 μm was used for the subsequent growth experiments. When the dislocation density is calculated, it seems to show that the larger the graphene area is, the smaller the dislocation density is. Therefore, what was the basis for the authors' choice of graphene of 2 μm width?

Answer. Graphene mask width of 2 um was chosen to have quick coalescence of GaN seeds, and to eliminate the need of use of special MOVPE conditions to promote lateral growth of GaN, such as Mg doping, high V/III ratio with NH3 flow interruptions [B.Beaumont et al., Phys. Status Solidi B 227, 1 (2001)]. Our focus was to investigate the feasibility of graphene as ELO mask.